# High Per Parameter: A Large-Scale Study of Hyperparameter Tuning for Machine Learning Algorithms

Moshe Sipper [ID]

Department of Computer Science, Ben-Gurion University, Beer-Sheva 8410501, Israel; sipper@bgu.ac.il

**Abstract:** Hyperparameters in machine learning (ML) have received a fair amount of attention, and hyperparameter tuning has come to be regarded as an important step in the ML pipeline. However, just how useful is said tuning? While smaller-scale experiments have been previously conducted, herein we carry out a large-scale investigation, specifically one involving 26 ML algorithms, 250 datasets (regression and both binary and multinomial classification), 6 score metrics, and 28,857,600 algorithm runs. Analyzing the results we conclude that for many ML algorithms, we should not expect considerable gains from hyperparameter tuning on average; however, there may be some datasets for which default hyperparameters perform poorly, especially for some algorithms. By defining a single *hp_score* value, which combines an algorithm's accumulated statistics, we are able to rank the 26 ML algorithms from those expected to gain the most from hyperparameter tuning to those expected to gain the least. We believe such a study shall serve ML practitioners at large.

**Keywords:** machine learning; hyperparemeters

## 1. Introduction

In machine learning (ML), a hyperparameter is a parameter whose value is given by the user and used to control the learning process. This is in contrast to other parameters, whose values are obtained algorithmically via training.

Hyperparameter tuning, or optimization, is often costly and software packages invariably provide hyperparameter defaults. Practitioners will often tune these—either manually or through some automated process—to gain better performance. They may resort to previously reported "good" values or perform some hyperparameter-tuning experiments.

In recent years, there has been increased interest in software that performs automated hyperparameter tuning, such as Hyperopt [1] and Optuna [2]. The latter, for example, is a state-of-the-art hyperparameter tuner which formulates the hyperparameter optimization problem as a process of minimizing or maximizing an objective function that takes a set of hyperparameters as an input and returns its (validation) score. It also provides pruning, i.e., automatic early stopping of unpromising trials. Moreover, our experience has shown it to be fairly easy to set up, and indeed we used it successfully in our research [3,4].

A number of recent works, which we shall review, have tried to assess the importance of hyperparameters through experimentation. We propose herein to examine the issue of hyperparameter tuning through a significantly more extensive empirical study than has been performed to date, involving multitudinous algorithms, datasets, metrics, and hyperparameters. Our aim is to assess just how much of a performance gain can be had per algorithm by employing a performant tuning method.

The next section presents an account of relevant previous work. Section 3 describes the experimental setup, followed by results in Section 4. We discuss our findings in Section 5, and end with concluding remarks in Section 6.

## 2. Previous Work

There has been a fair amount of work on hyperparameters and it is beyond this paper's scope to provide a detailed review. For that, we refer the reader to the recent

comprehensive review: "Hyperparameter Optimization: Foundations, Algorithms, Best Practices and Open Challenges" [5].

Interestingly, Ref. [5] wrote that "we would like to tune as few HPs [hyperparameters] as possible. If no prior knowledge from earlier experiments or expert knowledge exists, it is common practice to leave other HPs at their software default values...".

Ref. [5] also noted that "more sophisticated HPO [hyperparameter optimization] approaches in particular are not as widely used as they could (or should) be in practice" (the paper does not include an empirical study). We shall use a sophisticated HPO approach herein.

We present below only recent papers that are directly relevant to ours, "ancestors" of the current study, as it were.

A major work by [6] formalized the problem of hyperparameter tuning from a statistical point of view, defined data-based defaults, and suggested general measures quantifying the tunability of hyperparameters. The overall tunability of an ML algorithm or that of a specific hyperparameter was essentially defined by comparing the gain attained through tuning with some baseline performance, usually attained when using default hyperparameters. They also conducted an empirical study involving 38 binary classification datasets from OpenML, and six ML algorithms: elastic net, decision tree, k-nearest neighbors, support vector machine, random forest, and xgboost. Tuning was performed through a random search. They found that some algorithms benefited from tuning more than others, with elastic net and svm showing the highest improvement and random forest showing the lowest.

Ref. [7] presented a methodology to determine the importance of tuning a hyperparameter based on a non-inferiority test and tuning risk, i.e., the performance loss that is incurred when a hyperparameter is not tuned, but set to a default value. They performed an empirical study involving 59 datasets from OpenML and two ML algorithms: support vector machine and random forest. Tuning was performed through random search. Their results showed that leaving particular hyperparameters at their default value is noninferior to tuning these hyperparameters. In some cases, leaving the hyperparameter at its default value even outperformed tuning it.

Finally, Ref. [8] recently presented results and insights pertaining to the black-box optimization (BBO) challenge at NeurIPS 2020. Analyzing the performance of 65 submitted entries, they concluded that, "Bayesian optimization is superior to random search for machine learning hyperparameter tuning" (indeed this is the paper's title) (NB: a random search is usually better than a grid search, e.g., [9]). We shall use Bayesian optimization herein.

*The Current Study*

After examining these recent studies, we made the following decisions regarding the experiments that we shall carry out herein:

- Consider significantly more algorithms;
- Consider significantly more datasets;
- Consider Bayesian optimization, rather than weaker-performing random search or grid search.

## 3. Experimental Setup

Our setup involves numerous runs across a plethora of algorithms and datasets, comparing tuned and untuned performance over six distinct metrics. Below, we detail the following setup components:

1. Datasets;
2. Algorithms;
3. Metrics;
4. Hyperparameter tuning;
5. Overall flow.

### 3.1. Datasets.

We used the recently introduced PMLB repository [10], which includes 166 classification datasets and 122 regression datasets. As we were interested in performing numerous runs, we retained the 144 classification datasets with number of samples $\leq$ 10,992 and number of features $\leq$ 100, and the 106 regression datasets with number of samples $\leq$ 8192 and number of features $\leq$ 100. Figure 1 presents a summary of dataset characteristics. Note that classification problems are both binary and multinomial.

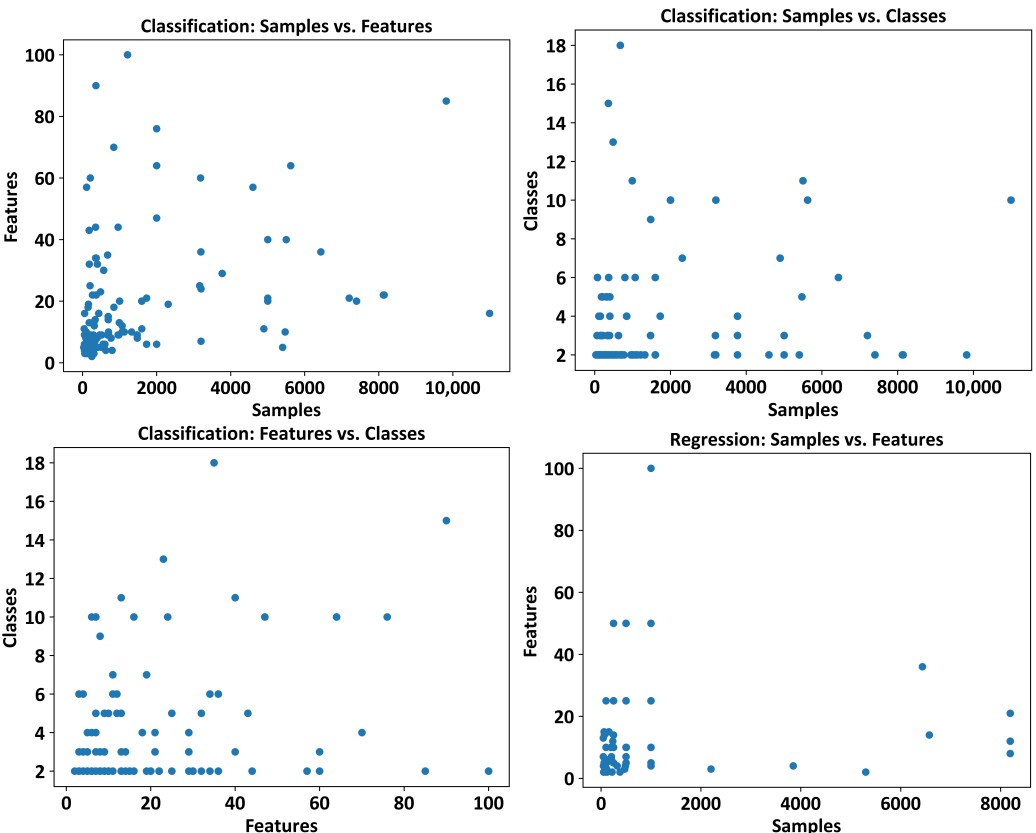

**Figure 1.** Characteristics of the 144 classification datasets and 106 regression datasets used in our study.

### 3.2. Algorithms

We investigated 26 ML algorithms—13 classifiers and 13 regressors—using the following software packages: scikit-learn [11], xgboost [12], and lightgbm [13]. The algorithms are listed in Table 1, along with the hyperparameter ranges or sets used in the hyperparameter search (described below).

### 3.3. Metrics

We used three separate metrics for classification problems:

1. Accuracy: a fraction of correct predictions ($\in [0, 1]$).
2. Balanced accuracy: an accuracy score that takes into account class imbalances, essentially the accuracy score with class-balanced sample weights [14] ($\in [0, 1]$).
3. F1 score: a harmonic mean of precision and recall; in the multi-class case, this is the average of the F1 score per class with weighting ($\in [0, 1]$)

We used three separate metrics for regression problems:

1. $R^2$ score: an $R^2$ (coefficient of determination) regression score function ($\in [-\infty, 1]$).
2. Adjusted $R^2$ score: a modified version of the $R^2$ score that adjusts for the number of predictors in a regression model. It is defined as $1 - (1 - r2) * (n - 1)/(n - p - 1)$,

with $r2$ being the $R^2$ score, $n$ being the number of samples, and $p$ being the number of features ($\in [-\infty, 1]$).

3.  Complement RMSE: a complement of root mean squared error (RMSE), defined as $1 - \text{RMSE}$ ($\in [-\infty, 1]$). This has the same range as the previous two metrics.

**Table 1.** Value ranges or sets used by Optuna for hyperparameter tuning. For ease of reference, we use the function names of the respective software packages: scikit-learn, xgboost, and lightgbm. Values sampled from a range in the log domain are marked as 'log', otherwise sampling is linear (uniform).

| Classification | | |
|---|---|---|
| **Algorithm** | **Hyperparameter** | **Values** |
| AdaBoostClassifier | n_estimators | [10, 1000] (log) |
| | learning_rate | [0.1, 10] (log) |
| DecisionTreeClassifier | max_depth | [2, 10] |
| | min_impurity_decrease | [0.0, 0.5] |
| | criterion | {gini, entropy} |
| GradientBoostingClassifier | n_estimators | [10, 1000] (log) |
| | learning_rate | [0.01, 0.3] |
| | subsample | [0.1, 1] |
| KNeighborsClassifier | weights | {uniform, distance} |
| | algorithm | {auto, ball_tree, kd_tree, brute} |
| | n_neighbors | [2, 20] |
| LGBMClassifier | n_estimators | [10, 1000] (log) |
| | learning_rate | [0.01, 0.2] |
| | bagging_fraction | [0.5, 0.95] |
| LinearSVC | max_iter | [10, 10,000] (log) |
| | tol | [$1 \times 10^{-5}$, 0.1] (log) |
| | C | [0.01, 10] (log) |
| LogisticRegression | penalty | {l1, l2} |
| | solver | {liblinear, saga} |
| MultinomialNB | alpha | [0.01, 10] (log) |
| | fit_prior | {True, False} |
| PassiveAggressiveClassifier | C | [0.01, 10] (log) |
| | fit_intercept | {True, False} |
| | max_iter | [10, 1000] (log) |
| RandomForestClassifier | n_estimators | [10, 1000] (log) |
| | min_weight_fraction_leaf | [0.0, 0.5] |
| | max_features | {auto, sqrt, log2} |
| RidgeClassifier | solver | {auto, svd, cholesky, lsqr, sparse_cg, sag, saga} |
| | alpha | [0.001, 10] (log) |
| SGDClassifier | penalty | {l2, l1, elasticnet} |
| | alpha | [$1 \times 10^{-5}$, 1] (log) |
| XGBClassifier | n_estimators | [10, 1000] (log) |
| | learning_rate | [0.01, 0.2] |
| | gamma | [0.0, 0.4] |
| **Regression** | | |
| **Algorithm** | **Hyperparameter** | **Values** |
| AdaBoostRegressor | n_estimators | [10, 1000] (log) |
| | learning_rate | [0.1, 10] (log) |

**Table 1.** *Cont.*

| Regression | | |
|---|---|---|
| **Algorithm** | **Hyperparameter** | **Values** |
| BayesianRidge | n_iter<br>alpha_1<br>lambda_1<br>tol | [10, 1000] (log)<br>$[1 \times 10^{-7}, 1 \times 10^{-5}]$ (log)<br>$[1 \times 10^{-7}, 1 \times 10^{-5}]$ (log)<br>$[1 \times 10^{-5}, 0.1]$ (log) |
| DecisionTreeRegressor | max_depth<br>min_impurity_decrease<br>criterion | [2, 10]<br>[0.0, 0.5]<br>{squared_error, friedman_mse, absolute_error} |
| GradientBoostingRegressor | n_estimators<br>learning_rate<br>subsample | [10, 1000] (log)<br>[0.01, 0.3]<br>[0.1, 1] |
| KNeighborsRegressor | weights<br>algorithm<br>n_neighbors | {uniform, distance}<br>{auto, ball_tree, kd_tree, brute}<br>[2, 20] |
| KernelRidge | kernel<br>alpha<br>gamma | {linear, poly, rbf, sigmoid}<br>[0.1, 10] (log)<br>[0.1, 10] (log) |
| LGBMRegressor | lambda_l1<br>lambda_l2<br>num_leaves | $[1 \times 10^{-8}, 10.0]$ (log)<br>$[1 \times 10^{-8}, 10.0]$ (log)<br>[2, 256] |
| LinearRegression | fit_intercept<br>normalize | {True, False}<br>{True, False} |
| LinearSVR | loss<br>tol<br>C | {epsilon_insensitive,<br>squared_epsilon_insensitive}<br>$[1 \times 10^{-5}, 0.1]$ (log)<br>[0.01, 10] (log) |
| PassiveAggressiveRegressor | C<br>fit_intercept<br>max_iter | [0.01, 10] (log)<br>{True, False}<br>[10, 1000] (log) |
| RandomForestRegressor | n_estimators<br>min_weight_fraction_leaf<br>max_features | [10, 1000] (log)<br>[0.0, 0.5]<br>{auto, sqrt, log2} |
| SGDRegressor | alpha<br>penalty | $[1 \times 10^{-5}, 1]$ (log)<br>{l2, l1, elasticnet} |
| XGBRegressor | n_estimators<br>learning_rate<br>gamma | [10, 1000] (log)<br>[0.01, 0.2]<br>[0.0, 0.4] |

*3.4. Hyperparameter Tuning*

For hyperparameter tuning, we used Optuna, a state-of-the-art automatic hyperparameter optimization software framework [2]. Optuna offers a define-by-run-style user API where one can dynamically construct the search space, and an efficient sampling algorithm and pruning algorithm. Moreover, our experience has shown it to be fairly easy to set up. Optuna formulates the hyperparameter optimization problem as a process of minimizing or maximizing an objective function that takes a set of hyperparameters as an input and returns its (validation) score. We used the default tree-structured Parzen estimator (TPE) Bayesian sampling algorithm. Optuna also provides pruning, i.e., the automatic early stopping of unpromising trials [2].

*3.5. Overall Flow*

Algorithm 1 presents the top-level flow of the experimental setup. For each combination of algorithm and dataset, we perform 30 replicate runs. Each replicate separately assesses model performance over the respective three classification or regression metrics. A replicate begins by splitting the dataset into training and test sets, and scaling them. Then, for each metric:

1.  Optuna is run over the training set for 50 trials to tune the model's hyperparameters, the best model is retained, and the best model's test-set metric score is computed.
2.  Fifty models are evaluated over the training set with default parameters, the best model is retained, and the best model's test-set metric score is computed. Strictly speaking, a few algorithms—decision tree, KNN, Bayesian—are essentially deterministic. For consistency, we still performed the 50 default hyperparameter trials. Further, our examination of the respective implementations revealed possible randomness, e.g., for decision tree, when *max_features* < *n_features*, the algorithm will select *max_features* at random; though the default is *max_features* = *n_features* we still took no chances of there being some hidden randomness deep within the code.

An evaluation of the model is carried out through five-fold cross-validation. At the end of each replicate, the test-set percent improvement in Optuna's best model is computed over the default's best model.

---

**Algorithm 1** Experimental setup (per algorithm and dataset)

---

**Input:**

    *algorithm* ← algorithm to run
    *dataset* ← dataset to be used
    *n_replicates* ← 30 (number of replicates)
    *n_trials* ← 50 (number of Optuna trials, also number of runs with default values)
    *time_limit* ← 72 h (for all replicates)

**Output:**

    Final scores (over test sets)

```
# 'metric1', 'metric2', 'metric3' are, respectively:
#      · For classification: accuracy, balanced accuracy, F1
#      · For regression: R², adjusted R², complement RMSE
# eval_score is 5-fold cross-validation score
```

 1: Load *dataset*
 2: **for** *rep* ← 1 to *n_replicates* **do**
 3:    Randomly split *dataset* into 70% *training_set* and 30% *test_set*
 4:    Fit `MinMaxScaler` to *training_set* and apply fitted scaler to *training_set* and *test_set*
 5:    **for** *metric* in 'metric1', 'metric2', 'metric3' **do**
 6:        Run Optuna with *algorithm* for *n_trials* trials over *training_set* and obtain *best_model*  # use *eval_score* for single-trial evaluation
 7:        Train *best_model* over *training_set*
 8:        Compute *metric* for *best_model* over *test_set*
 9:        **for** *i* in *n_trials* **do**
10:           Initialize a *model* with default hyperparameters
11:           Evaluate *model* over *training_set* using *eval_score*
12:           **if** *eval_score* is best obtained so far **then**
13:               Save *model* as *best_model*
14:        Train *best_model* over *training_set*
15:        Compute *metric* for *best_model* over *test_set*
16:    *imp1*, *imp2*, *imp3* = percent improvement Optuna over default for 'metric1', 'metric2', 'metric3'  # Compute and record replicate scores
17:    **if** runtime > *time_limit* **then**
18:        break

---

## 4. Results

A total of 96,192 replicates were performed, each comprising 300 algorithm runs (3 metrics × 50 Optuna trials, 3 metrics × 50 default trials), with the final tally thus being

28,857,600 algorithm runs. Note that for each run, we used the `fit` method of the respective algorithm five times during five-fold cross validation, i.e., the learning algorithm was executed five times. Table 2 presents our results.

**Table 2.** Compendium of final results over 26 ML algorithms, 250 datasets, 96,192 replicates, and 28,857,600 algorithm runs. A table row presents results of a single ML algorithm, showing a summary of all replicates and datasets. A table cell summarizes the results of an algorithm–metric pair. Cell values show *median* and *mean(std)*, where *median*: median over all replicates and datasets of Optuna's percent improvement over default; *mean(std)*: mean (with standard deviation) over all replicates and datasets of Optuna's percent improvement over default. The total number of replicates for which these statistics were computed is also shown. Acc: accuracy score; Bal: balanced accuracy score; F1: F1 score; $R^2$: $R^2$ score; Adj $R^2$: adjusted $R^2$ score; C-RMSE: complement RMSE; Reps: total number of replicates [†].

| Classification | | | | | | | |
|---|---|---|---|---|---|---|---|
| **Algorithm** | **Acc** | | **Bal** | | **F1** | | **Reps** |
| | *Median* | *Mean(std)* | *Median* | *Mean(std)* | *Median* | *Mean(std)* | |
| AdaBoostClassifier | 1.9 | 20.9 (65.3) | 2.2 | 21.5 (57.4) | 1.9 | 39.3 (150.1) | 4320 |
| DecisionTreeClassifier | 0.0 | 115.6 ($2.3 \times 10^3$) | 0.0 | 96.0 ($2.2 \times 10^3$) | 0.0 | 55.7 ($1.3 \times 10^3$) | 4220 |
| GradientBoostingClassifier | 0.5 | 45.1 ($1.4 \times 10^3$) | 0.6 | 48.9 ($1.4 \times 10^3$) | 0.6 | 42.0 ($1.1 \times 10^3$) | 4096 |
| KNeighborsClassifier | 0.8 | 3.8 (13.8) | 1.8 | 5.9 (16.7) | 1.5 | 5.1 (17.7) | 4254 |
| LGBMClassifier | 0.0 | 1.2 (11.5) | 0.0 | 1.0 (11.9) | 0.0 | 0.9 (12.1) | 4287 |
| LinearSVC | 0.0 | 1.0 (8.3) | 0.0 | 1.9 (8.5) | 0.0 | 1.7 (8.2) | 4299 |
| LogisticRegression | 0.0 | 1.5 (8.2) | 0.0 | 3.4 (12.5) | 0.0 | 3.4 (11.8) | 4307 |
| MultinomialNB | 0.0 | 9.8 (58.9) | 8.5 | 27.5 (48.9) | 10.5 | 40.5 (128.6) | 4149 |
| PassiveAggressiveClassifier | 1.9 | 7.8 (24.0) | 1.8 | 5.9 (18.6) | 3.0 | 10.8 (28.8) | 4301 |
| RandomForestClassifier | 0.0 | 153.6 ($2.3 \times 10^3$) | 0.0 | 218.8 ($3.0 \times 10^3$) | 0.0 | 134.1 ($2.0 \times 10^3$) | 4320 |
| RidgeClassifier | 0.0 | 1.0 (6.8) | 0.0 | 1.4 (7.3) | 0.0 | 1.9 (7.9) | 4273 |
| SGDClassifier | 1.2 | 5.0 (20.6) | 1.6 | 5.2 (16.7) | 2.0 | 8.6 (26.7) | 4212 |
| XGBClassifier | 0.0 | 13.5 (643.1) | 0.0 | 11.4 (431.2) | 0.0 | 10.1 (467.6) | 4111 |
| Regression | | | | | | | |
| **Algorithm** | **$R^2$** | | **Adj** | | **C-RMSE** | | **Reps** |
| | *median* | *mean(std)* | *median* | *mean(std)* | *median* | *mean(std)* | |
| AdaBoostRegressor | 2.0 | 3.6 (33.5) | 2.1 | −741.3 ($9.5 \times 10^3$) | 3.8 | 5.1 (20.9) | 3179 |
| BayesianRidge | 0.0 | $6.8 \times 10^3$ ($3.7 \times 10^5$) | −0.0 | −3.3 (55.3) | 0.0 | 1.0 (9.8) | 3117 |
| DecisionTreeRegressor | 3.8 | 61.5 (788.0) | 4.0 | 49.1 (841.5) | 7.0 | 63.4 ($1.3 \times 10^3$) | 3150 |
| GradientBoostingRegressor | 1.6 | 17.3 (430.9) | 1.7 | −6.6 $\times 10^5$ ($2.4 \times 10^7$) | 4.1 | 2.3 (126.4) | 3180 |
| KNeighborsRegressor | 3.5 | 77.8 (627.4) | 3.5 | 18.8 (471.6) | 4.5 | 203.5 ($5.3 \times 10^3$) | 3160 |
| KernelRidge | 69.5 | −9.3 $\times 10^5$ ($5.0 \times 10^7$) | 65.9 | $3.6 \times 10^3$ ($1.7 \times 10^5$) | 49.5 | $1.7 \times 10^3$ ($8.1 \times 10^4$) | 3053 |
| LGBMRegressor | 0.0 | 0.0 (25.6) | 0.0 | −1.2 (34.4) | 0.0 | 0.4 (2.1) | 3179 |
| LinearRegression | 0.0 | 2.3 (70.4) | 0.0 | −35.1 (469.4) | 0.0 | −1.7 (62.8) | 3170 |
| LinearSVR | 25.1 | 86.4 ($2.7 \times 10^3$) | 24.3 | 173.5 ($2.8 \times 10^3$) | 23.9 | 159.7 ($2.2 \times 10^3$) | 3161 |
| PassiveAggressiveRegressor | 71.6 | 180.7 ($1.7 \times 10^3$) | 58.5 | −304.3 ($4.1 \times 10^3$) | 62.0 | 331.9 ($5.5 \times 10^3$) | 3167 |
| RandomForestRegressor | −0.1 | 1.5 (44.2) | −0.2 | −1.2 $\times 10^3$ ($4.6 \times 10^4$) | −0.5 | −1.5 (13.2) | 3180 |
| SGDRegressor | 0.0 | 2.6 (68.6) | 0.0 | −41.4 ($2.0 \times 10^3$) | 0.0 | 2.2 (39.8) | 3167 |
| XGBRegressor | 0.9 | 20.0 (717.1) | 0.8 | −675.6 ($7.4 \times 10^3$) | 2.3 | 6.8 (164.6) | 3180 |

[†] The number of replicates may be smaller than the maximal possible value ($144 \times 30 = 4320$ for classification datasets, and $106 \times 30 = 3180$ for regression datasets). This is due to edge cases that cause a single replicate to terminate with an error, the vicissitudes of life on the cluster, and (in small part) long runtimes evoking the 72 h timeout (this happened with GradientBoostingClassifier for 14 datasets and with XGBClassifier for 8 datasets).

Table 2 shows several interesting points. First, regressors are somewhat more suscepti-ble to hyperparameter tuning, i.e., there is more to be gained by tuning vis-a-vis the default hyperparameters.

For most classifiers and—to a lesser extent—regressors, the median value shows little to be gained from tuning, yet the mean value along with the standard deviation suggests that for some algorithms there is a wide range in terms of tuning effectiveness. Indeed, by examining the collected raw experimental results, we noted that there was a "low-hanging fruit" case at times. The default hyperparameters yielded very poor performance on some datasets, leaving room for considerable improvement through tuning.

It would seem useful to define a "bottom-line" measure—a summary score, as it were, which essentially summarizes an entire table row, i.e., an ML algorithm's sensitivity to hyperparameter tuning. We believe any such measure would be inherently arbitrary to some extent; that said, we nonetheless put forward the following definition of *hp_score*:

- The 13 algorithms and 9 measures of Table 2 are considered (separately for classi-fiers and regressors) as a dataset with 13 samples and the following 9 features: *metric1_median*, *metric2_median*, *metric3_median*, *metric1_mean*, *metric2_mean*, *metric3_mean*, *metric1_std*, *metric2_std*, *metric3_std*.
- Scikit-learn's RobustScaler is applied, which scales features using statistics that are robust to outliers: "This Scaler removes the median and scales the data according to the quantile range (defaults to IQR: Interquartile Range). The IQR is the range between the 1st quartile (25th quantile) and the 3rd quartile (75th quantile). Centering and scaling happen independently on each feature..." [14].
- The *hp_score* of an algorithm is then simply the mean of its nine scaled features.

This *hp_score* is unbounded because improvements or impairments can be arbitrarily high or low. A higher value means that the algorithm is expected to gain more from hyperparameter tuning, while a lower value means that the algorithm is expected to gain less from hyperparameter tuning (on average).

Table 3 presents the *hp_score*s of all 26 algorithms, sorted from highest to lowest per algorithm category (classifier or regressor). While simple and immanently imperfect, *hp_score* nonetheless seems to summarize the trends observable in Table 2 fairly well.

**Table 3.** The *hp_score* of each ML algorithm, computed from the values in Table 2. A higher value means that the algorithm is expected to gain more from hyperparameter tuning, while a lower value means that the algorithm is expected to gain less from hyperparameter tuning.

| Classification | | Regression | |
| --- | --- | --- | --- |
| RandomForestClassifier | 3.89 | KernelRidge | 2110.75 |
| DecisionTreeClassifier | 2.43 | GradientBoostingRegressor | 183.97 |
| GradientBoostingClassifier | 1.52 | BayesianRidge | 35.19 |
| MultinomialNB | 1.36 | PassiveAggressiveRegressor | 5.34 |
| AdaBoostClassifier | 0.79 | LinearSVR | 2.13 |
| PassiveAggressiveClassifier | 0.56 | KNeighborsRegressor | 0.59 |
| XGBClassifier | 0.38 | DecisionTreeRegressor | 0.35 |
| SGDClassifier | 0.35 | RandomForestRegressor | 0.09 |
| KNeighborsClassifier | 0.27 | AdaBoostRegressor | −0.07 |
| LogisticRegression | −0.08 | XGBRegressor | −0.10 |
| LinearSVC | −0.09 | SGDRegressor | −0.23 |
| RidgeClassifier | −0.10 | LinearRegression | −0.25 |
| LGBMClassifier | −0.10 | LGBMRegressor | −0.26 |

## 5. Discussion

The main takeaway from Tables 2 and 3 is as follows. For most ML algorithms, we should not expect huge gains from hyperparameter tuning *on average*; however, there may be some datasets for which default hyperparameters perform poorly, especially for some algorithms. In particular, those algorithms at the bottom of the lists in Table 3 would

likely not benefit greatly from a significant investment in hyperparameter tuning. Some algorithms are robust to hyperparameter selection, while others are somewhat less robust.

Perhaps the main limitation of this work (as in others involving hyperparameter experimentation) pertains to the somewhat subjective choice of value ranges (Table 1). This is, ipso facto, unavoidable in empirical research such as this. While this limitation cannot be completely overcome, it can be offset given that the code is publicly available at https://github.com/moshesipper (accessed on 1 September 2022), and we and others may enhance our experiment and add additional findings. Indeed, we hope this to be the case.

Table 3 can be used in practice by an ML practitioner to:

1. Decide how much to invest in hyperparameter tuning of a particular algorithm;
2. Select algorithms that require less tuning to hopefully save time—as well as energy [15].

## 6. Concluding Remarks

We performed a large-scale experiment of hyperparameter-tuning effectiveness, across multiple ML algorithms and datasets. We found that for many ML algorithms, we should not expect considerable gains from hyperparameter tuning *on average*; however, there may be some datasets for which default hyperparameters perform poorly, especially for some algorithms. By defining a single *hp_score* value, which combines an algorithm's accumulated statistics, we were able to rank the 26 ML algorithms from those expected to gain the most from hyperparameter tuning to those expected to gain the least. We believe such a study may serve ML practitioners at large, in several ways, as noted above.

There are many avenues for future work:

1. Algorithms may be added to the study.
2. Datasets may be added to the study.
3. Hyperparameters that have not been considered herein may be added.
4. Specific components of the setup may be managed (e.g., the metrics and the scaler of Algorithm 1).
5. Additional summary scores, like the *hp_score*, may be devised.
6. For algorithms at the top of the lists in Table 3, we may inquire as to whether particular hyperparameters are the root cause of their hyperparameter sensitivity; further, we may seek out better defaults. For example, [16] recently focused on hyperparameter tuning for KernelRidge, which is at the top of the regressor list in Table 3. Ref [6] discussed the tunability of a specific hyperparameter, though they noted the problem of hyperparameter dependency.

Given the findings herein, it seems that, more often than not, hyperparameter tuning will not provide huge gains over the default hyperparameters of the respective software packages examined. A modicum of tuning would seem to be advisable, though other factors will likely play a stronger role in final model performance, including, to name a few, the quality of raw data, the solidity of data preprocessing, and the choice of ML algorithm (curiously, the latter can be considered a tunable hyperparameter [3]).

**Funding:** This research received no external funding.

**Data Availability Statement:** Not applicable.

**Acknowledgments:** I thank Raz Lapid for helpful comments.

**Conflicts of Interest:** The authors declare no conflicts of interest.

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
