# Peer review of "High Per Parameter: A Large-Scale Study of Hyperparameter Tuning for Machine Learning Algorithms"

_algorithms, doi:10.3390/a15090315_

Round 1

Reviewer 1 Report

The introduction is too short and does not provide motivation nor reasoning for conducting study on the topic in the first place. Critical analysis of papers which constitute previous work is rather scarce. What is missing in recent and relevant studies in the field and how (with much more details) is this resolved and addressed in this paper? There is a lack of discussion on presented findings and is therefore unclear what are major contributions of this paper and how they add to the extant body of knowledge. The authors also failed to explain in detail what are implications for researchers and practitioners and what are limitation of the study being carried out. The last section should also be enhanced in the manner that is clear to the reader what conclusions have been drawn and what future work will focus on. The authors are also advised not to put so much information in table headings as it looks cluttered. Considering all the aforementioned, the authors are advised to rework their paper.

Author Response

Thank you for taking the time to read my paper and suggest improvements. 

I have improved the introduction.

As for previous work -- I've noted that "There has been a fair amount of work on hyperparameters and it is beyond this paper’s scope to provide a review." Indeed, I went on to cite a major 70-page review from a few months ago. I tried (and following the comments -- retried), to seek out those papers that are direct "ancestors", as it were, of the current one. That is, papers involving experimentation along the lines I'm doing in the paper. I ended this section with a new subsection describing the "hole" I'm attempting to fill. 

I agree about table headings. Table 2 in particular is perhaps somewhat lengthy in its caption, but in my experience it helps readers in making tables more self-contained, without the need to delve back-and-forth between table and main text (indeed, I've been advised in the past by editors to put MORE into my captions...). That said, I've shortened Table 2's caption.

I moved the discussion to a separate section.

I added implications, limitations, and conclusions, as suggested, in the two last sections.

Reviewer 2 Report

This study performed a large-scale experiment of hyperparameter-tuning effectiveness,  across multiple ML algorithms and datasets. From my view, this paper is not well organized and the proposed method is not valuable for this research filed. After reviewed this paper, there are some questions and suggestions as follows.

  1. What are the advantages and disadvantages of this study compared to the existing studies in this area? This needs to be addressed explicitly and in a separate subsection.
  2. There are many grammatical mistakes and typo errors.
  3. You must review all significant similar works that have been done. Also, review some of the good recent works that have been done in this area and are more similar to your paper. 
  4. The experimental results indicate that they perform well, but providing a stronger theoretical analysis and justification for the algorithm would be more convincing. To clearly state the objective of the research in terms of problems to address and expected results and show how the proposed technique will advance the state of the art by overcoming the limitations of the existing work. Also, the results obtained must be interpreted.
  5. Some final cosmetic comments:

* The results of your comparative study should be discussed in-depth and with more insightful comments on the behaviour on various case studies. Discussing results should not mean reading out the tables and figures once again.
* Avoid lumping references as in [x, y] and all other. Instead summarize the main contribution of each referenced paper in a separate sentence. For scientific and research papers, it is not necessary to give several references that say exactly the same. Anyway, that would be strange, since then what is innovative scientific contribution of referenced papers? For each thesis state only one reference.
* Avoid using first person.
* Avoid using abbreviations and acronyms in title, abstract, headings and highlights.
* Please avoid having heading after heading with nothing in between, either merge your headings or provide a small paragraph in between.
* The first time you use an acronym in the text, please write the full name and the acronym in parenthesis. Do not use acronyms in the title, abstract, chapter headings and highlights.
* The results should be further elaborated to show how they could be used for the real applications.

* Are all the images used in this work copyrights free? If not, have the authors obtained proper copyrights permission to re-use them? Please kindly clarify, and this is just to ensure all the figures are fine to be published in this work.

* Also, the list of references should be carefully checked to ensure consistency with between all references and their compliances with the journal policy on referencing.

Author Response

Thank you for taking the time to read my paper and suggest improvements. 

There is now a separate subsection comparing this study to previous ones.

I have carefully reread the paper and corrected grammatical mistakes.

As for previous work -- I've noted that "There has been a fair amount of work on hyperparameters and it is beyond this paper’s scope to provide a review." Indeed, I went on to cite a major 70-page review from a few months ago. I tried (and following the comments -- retried), to seek out those papers that are direct "ancestors", as it were, of the current one. That is, papers involves experimentation along the lines I'm doing in the paper. I ended this section with the "hole" I'm attempting to fill -- the new subsection detailing the paper's added value.

Removed first person.

Fixed abbreviations and acronyms, first use is written parenthesis, as suggested.

I moved the discussion to a separate section.

Fixed headings.

I added implications, limitations, and conclusions, as suggested, in the two last sections.

Summarized contribution of each paper, as suggested.

Images are all mine, so no problem with copyright.

I've double-checked the list of references.

Reviewer 3 Report

Summary: In this paper, the authors studied the hyperparameters tuning problem in machine learning (ML). Specifically, the authors conducted a large-scale investigation involving 26 ML algorithms, 250 datasets, 6 score metrics, and 28,857,600 algorithm runs. The results conclude that for many ML algorithms, hyperparameter tuning does not give considerable gains on average. 

Comments:

1 . My major concern of this paper is the novelty part. This work seems to present a comprehensive investigation on the empirical effectiveness of hyperparameter tuning via tons of experiments. Yet I feel the final conclusion is not really surprising, or give any useful new information to the readers (such as how to better train the model via your findings?) This is more like a combination of experiments.

2. Although the authors conducted tons of experiments on hyperparameter tuning, these experiments are mostly regarding traditional ML methods such as tree-based or linear models. Such models by natural does not contain too many hyperparameters and the since the model is relatively simple, the effect of hyperparameters are rather limited. Therefore, it is questionable whether the conclusion would still holds for deep learning models, which have much larger hyperparater space and more complicated decision boundaries. I think the authors should also consider those settings for a fair and comprehensive conclusion.

Author Response

Thank you for taking the time to read my paper and suggest improvements. 

I've added the exact "hole" I'm filling in the new subsection at the end of Prev Work. I also added implications, limitations, and conclusions, as suggested, in the two last sections.

I agree re. deep learning -- but this study is about ML alone. I have seen much use of ML (including by myself) and I very much hope my study can help ML practitioners. 

Round 2

Reviewer 1 Report

Considering that the authors failed to address my concerns to the sufficient extent, I would not argue for the acceptance of this paper.

Reviewer 2 Report

Good revisions have been made in the paper and the revised version has the necessary qualities for acceptance compared to the previous version. In my opinion, the article is acceptable in its current form.

Reviewer 3 Report

I thank the authors for their responses. The authors argue that "this study is about ML alone”, which is a bit confusing as deep learning is also part of ML and this largely limits the practicality and usefulness of the study, given that the novelty of this work is already insufficient. Therefore, I would recommend reject for this paper.